# Energy Recovery from Waste Paper and Deinking Sludge to Support the Demand of the Paper Industry: A Numerical Analysis

Simona Di Fraia [ID] and M. Rakib Uddin *

Department of Engineering, University of Naples "Parthenope", 80143 Napoli, Italy;
simona.difraia@uniparthenope.it
* Correspondence: mohammadrakib.uddin001@studenti.uniparthenope.it

**Abstract:** The recovery of fibres from waste paper (WP) and deinking sludge (DIS) reduces the stress on nature compared to the collection of virgin pulp for paper production. Moreover, if not recycled, WP and DIS are mainly landfilled and incinerated, being thus responsible for the release of greenhouse gases (GHGs) into the atmosphere. In this context, energy recovery from WP and DIS would contribute to increasing energy independence and improving waste management in the pulp industry. From a broader perspective, it would increase renewable energy generation, supporting the paper industry in reducing fossil fuel consumption and GHGs emissions, in line with the goals of the European Union (EU) Green Deal 2021. For these reasons, in the present study, the combined heat and power generation potentiality of WP–DIS blends through gasification in combination with an internal combustion engine is numerically assessed for the first time. The air gasification process is simulated by applying a restricted chemical equilibrium approach to identify the optimum operating temperature (850 °C) and equivalence ratio (0.2). Electrical and thermal energy generation potentiality, considering WP and DIS production in the EU in 2019, is estimated to be in the ranges of 32,950–35,700 GWh and 52,190–56,100 GWh, respectively. Thus, it can support between 25 and 28% of the electrical and 44–48% of the thermal energy demand of the paper manufacturing sector, reducing the $CO_2$ emission in the range of 24.8–28.9 Gt.

**Keywords:** waste paper; deinking sludge; pellet; gasification; restricted chemical equilibrium model; syngas; sensitivity analysis; optimization; combined heat and power

## 1. Introduction

The pulp and paper industry presents a significant energy demand, mainly satisfied by fossil fuels [1,2], which cause high greenhouse gases (GHGs) and pollutant emissions [3]. Moreover, the pulp and paper industry generates a considerable quantity of waste as a small fraction of starting biomass is transformed into the final products of paper [1]. Thus, this sector could increase renewable energy production by recovering its waste, contributing to the sustainable energy transition. Another way to increase the eco-efficiency of this sector is recycling used paper to extract the pulp. Indeed, such a practice is globally increasing to reduce the stress on nature to collect virgin pulp from trees and the environmental pollution caused by landfilling or incinerating the used paper [4–6]. It is estimated that pulp collection from recycled used paper could save on average 8.2 million trees annually [7]. Cellulosic fibre length, as well as strength, decreases during the collection of pulp, requiring its blending with virgin fibres for paper production. However, used paper can be recycled three to eight times to collect cellulosic fibres, then the used paper has to be discarded as solid waste [8,9]. Based on the data presented in the CEPI-2020 (Confederation of European Paper Industries) report, in 2019, world average used paper recycling was 58.6% with the highest rate in the European Union (EU) (72.5%), followed by North America (65.7%), Asia (53.9%), Latin America (47.2%), and Africa (35.2%) [4].

During the recycling of used paper to collect the pulp, waste paper (WP) and deinking sludge (DIS) are generated as waste products. WP consists of fibre lumps, staples, sand,

glass, and plastics. A wide variety of constituents is present in DIS, which are short fibres, chemicals used as coatings, and fillers employed during paper manufacturing from virgin or recycled pulp to improve the paper quality, such as kaolin ($Al_2O_3$, $SiO_2$), talc ($Mg_3Si_4O_{10}(OH)_2$), calcium carbonate ($CaCO_3$), clays, ink particles, extractive substances, and deinking additives applied for used paper recycling (e.g., $Na_2SiO_3$, $NaOH$, $H_2O_2$, $CaCl_2$, fatty acid, and fatty acid soap) [10,11]. The lower heating value (LHV) of WP and DIS fluctuates from 15.0 to 26.61 MJ/kg as dry solid (DS) and 4.0 to 7.57 MJ/kg as DS, respectively [6,11].

In 2019, new paper and board production in the EU was 75.8 Mt of which 57.5 Mt were recycled generating 24.3 Mt of WP [12]. The quantity of DIS generated during the recycling of used paper is between 20% (for newsprint) and 40% (for tissue paper) of WP and is expected to reach between 48 and 86%, respectively, in the next 50 years [13,14]. In 2019, DIS production in the EU was in the range of 4.86 to 9.72 Mt as DS. The most common practices for disposal of the WP and DIS generated during used paper recycling are landfilling and incineration, which are responsible for the release of GHGs of $CH_4$, NO, $N_2O$, $CO_2$, CO, $SO_2$, HCl, mercury, dioxins, furan, polychlorinated biphenyls, and polycyclic aromatic hydrocarbons to the environment [6,15–19]. However, many studies have considered energy recovery from municipal solid waste (MSW), whereas the potential benefits of WP and DIS appear to be underestimated [20] even though their valorisation could contribute to decreasing the energy consumption from fossil sources in the paper manufacturing sector. This, together with the avoided waste disposal, would reduce the GHGs emissions of such a sector.

Thermal treatment appears to be very promising compared to the biological method for energy recovery from WP and DIS due to:

- the higher conversion rate of carbonaceous content to energy product (more than 80% for thermal treatments whereas in the range of 30–60% for biological methods);
- the lower processing time (thermal treatments require 30 to 70 min, whereas biological treatment needs between seven and 105 days) [21–23].

Considering the more common thermal treatments used for energy recovery from biomass, gasification is characterized by:

- a high carbon conversion efficiency (CCE) (from 60 to 80%) and cold gas efficiency (CGE) (between 60 and 90%);
- that migration of heavy or toxic metals from fed materials to the product phase is negligible;
- the possibility to use the gaseous product as a fuel in internal combustion engines (ICE) or gas turbines or microgas turbines without any modification [17,24–30].

Gasification is a thermal treatment that converts the energy content of a biomass to a gaseous phase at a temperature higher than 700 °C and atmospheric pressure in oxygen-deficient conditions. The gaseous product formed during biomass gasification consists of $H_2$, $CH_4$, CO, $CO_2$, and other lighter hydrocarbons with tar content and is commonly designated as syngas. Due to the easier availability and low cost, the air is frequently used as a gasifying agent to supply oxygen in the gasification process. However, based on the specification of syngas properties, other gasifying agents like pure $O_2$, steam, $CO_2$, a mixture of air–steam, $O_2$–steam, and $CO_2$–steam, may be also used. LHV of syngas varies between 3.0 and 9.73 MJ/$Nm^3$ depending on the properties of biomass used for the gasification and operating conditions [22,24,25,31,32].

Syngas composition and LHV as well as the performance of the conversion process (CCE and CGE) during air-gasification of biomass depends on the quality of fed materials and operating parameters, such as temperature and equivalence ratio (ER) [24,25]. For the sake of completeness, ER is the ratio between the actual air to biomass weight fed to the gasifier to the stoichiometric air to biomass weight required for complete combustion [33].

The available studies on the gasification of WP and DIS are limited [11,34]. Air gasification of WP and DIS blends (95% WP and 5% DIS by weight) in a pilot-scale circulating

fluidized bed reactor (FBR) are analysed by Rivera et al. (2016) [11]. They obtain a syngas with a tar concentration of 11.44 g/Nm$^3$ and an LHV of 5 MJ/Nm$^3$, considering a process temperature of 850 °C and an ER of 0.3. An experimental campaign on co-gasification of WP–DIS pellets (consisting of 80% reject fibres and 20% mixed plastic by weight) with wood chips was carried out by Ouadi et al. (2013) [34]. Through 12 experimental tests, the authors identify optimum conditions: a blending ratio of WP–DIS pellets and wood chips of 80:20 by weight, a temperature of 1000 °C, and an ER of 0.22. The generated syngas presents a tar concentration of 5.8 g/Nm$^3$ and an LHV of 7.3 MJ/Nm$^3$.

Identifying optimum operating parameters for biomass gasification through experimental campaigns is time-consuming and costly as several tests, in the range of 10 to 23, have to be performed [33–38]. Using computer-aided simulations of the biomass gasification process based on experimental outcomes can significantly reduce the time and cost of predicting the optimum operating parameters and process performances. As an example, Aspen Plus software allows identifying the conditions to improve the plant design, process limitations, or even failure conditions, which can improve the profit of an existing or proposed production plant [39].

The analysis of the literature highlights that no works are available on numerical modelling of gasification of WP and DIS except for an article where a mixture of used paper discarded as MSW is analysed [40]. In such a paper, Safarian et al. (2019) [40] develop a model to simulate the gasification of mixed-used paper and validate it by considering experimental outcomes related to the gasification of wood presented in the literature, founding an acceptable agreement. The authors identify the optimum operating temperature (1000 °C) and ER (0.3) to maximize the concentration of H$_2$ and CO of syngas, which is characterized by an LHV of 4.62 MJ/Nm$^3$. Regarding the process performance, they observe a CGE of 70.6%.

Extending the literature review on biomass gasification highlights the significant interest of the scientific community in this topic. However, focusing on numerical simulation to assess the CHP generation potentiality of biomass through gasification integrated with an ICE, the available literature is limited [41–43]. Energy recovery from sewage sludge (SS) through gasification integrated with an ICE was assessed by Di Fraia et al. (2021) [41]. The authors develop a numerical model through the software Aspen Plus estimating the electrical (29.2%), thermal (45.92%), and cogeneration efficiencies (53.1%). The same biomass was analysed by Brachi et al. (2022) [42], which estimate electrical (19.3%) and thermal (48.7%) efficiencies for a similar energy recovery configuration. An integrated system composed of a gasifier and an ICE was investigated also by Villarini et al. (2019) [43], taking into account the energy valorisation of hazelnut shell and olive pruning, estimating an electrical efficiency of 30% and 26%, respectively, and a cogeneration efficiency of 64% and 41%.

Based on average global pulp and paper industries' energy demand data, the electrical and thermal energy required to produce paper from wood is estimated as 1.68 MWh/ton and 1.55 MWh/ton, respectively [1,44]. Energy recovery from WP and DIS through a gasifier integrated with an ICE could supply a fraction of the electrical and thermal energy demand for the paper production process. Therefore, in the present study, a numerical model is developed to assess the CHP generation potentiality of WP–DIS blends through gasification in combination with an ICE using Aspen Plus V8.8 software (Bedford, MA, USA), for the first time. The gasification model is developed by applying a restricted chemical equilibrium approach [24,41,42,45]. The model is calibrated against experimental data available in the literature related to the gasification of WP–DIS blends [11] and validated by considering the outcomes of an experimental campaign on bamboo chips gasification [46]. The developed model is used to predict the optimum operating parameters for gasification of WP–DIS pellets (made by mixing 85 wt% WP and 15 wt% DIS) by analysing the effect of process temperature and ER on syngas composition, process performances, and net power obtained from gasification products. Finally, electrical and thermal energy that could be generated by considering the proposed system to treat WP and DIS produced in the EU in

2019 is assessed, together with the achievable reduction of $CO_2$ emissions compared to the use of conventional fuels.

Therefore, the present article presents a sustainable and energy-efficient solution for the paper and pulp industry. Indeed, CHP from WP and DIS may contribute to:

- reduce the waste landfilling;
- increase the energy dependence of the paper and pulp industry;
- at the global level, increase renewable energy generation and reduce GHGs.

The paper is organized as follows. The numerical model developed together with modelling assumptions is described in Section 2. The input parameters of the analysed case study and the main results of the study, including those of the sensitivity analyses, are illustrated in Section 3. Finally, the main findings and future developments of the work are presented in Section 4.

## 2. Materials and Methods

Energy recovery as CHP from WP and DIS generated during recycling of pulp from used paper through gasification integrated with an ICE is numerically analysed. The numerical model proposed in this work is calibrated against experimental data on syngas generation from WP–DIS pellets consisting of 95 wt% WP and 5 wt% DIS in a pilot-scale FBR, characterized by 300 mm reactor diameter, 8764 mm height, and recirculation pipe diameter 127 mm where gasifying agent air completes the fluidization of the bed [11]. After calibration, the model is validated against the experimental outcomes of bamboo chips gasification in a laboratory scale fixed bed gasifier with 100 mm diameter and 1400 mm height [47]. Two distinct WP–DIS pellets are considered in the analysis, from now on M1 (95% WP and 5% DIS by weight) and M2 (85% WP and 15% DIS by weight). The detailed experimental procedure for WP and DIS collection, sample pellets preparation, and characterization as well as details on the experimental campaign, are available in the literature [11].

The numerical model to simulate the conversion of WP–DIS blends to CHP is developed in Aspen Plus V8.8. The software library does not have a unique block for either the gasification process or the ICE. Thus, gasification is modelled considering the processes from which it is composed: drying, pyrolysis, gasification, and partial combustion to transform the energy content present in the WP–DIS pellet to syngas [22,48]. As commonly proposed in the available literature, the ICE is simulated by considering four consecutive blocks, a compressor followed by a chemical reactor to complete the combustion at constant volume, a turbine, and finally, a heat exchanger for cooling at constant volume [41,43,49,50].

The process flowsheet to simulate the CHP generation from WP–DIS pellets is presented in Figure 1.

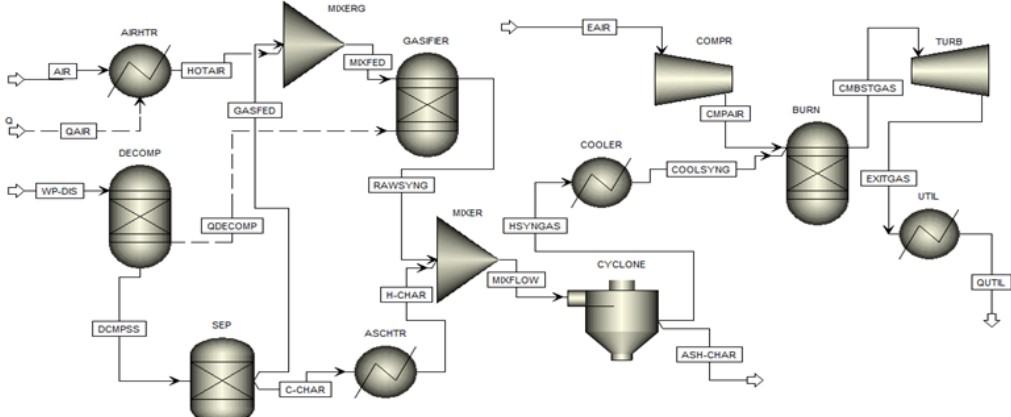

**Figure 1.** Flowsheet related to the simulation on CHP generation from WP–DIS pellets.

The first two processes, drying, and pyrolysis of the fed stream (WP–DIS), are completed in an RYield reactor (DECOMP). Such a block decomposes the non-conventional stream of WP–DIS pellet to conventional (C, $H_2$, $N_2$, $Cl_2$, $F_2$, S, and $H_2O$) and non-conventional (ash) components, based on ultimate elemental analysis implemented through a FORTRAN subroutine in a calculator. The temperature of the DECOMP block is set at 400 °C, which is indicated in the pertinent literature as the optimum temperature for pyrolysis of biomass [31]. The product exiting from DECOMP block (DECMPSS) is separated in a separator (SEP) into two sub-streams: a fraction of carbon that participates in gasification reactions (GASFED) and the remaining that forms char and ash (C-CHAR). The gasification fed stream (GASFED) is mixed in a mixer (MIXERG) with air (HOTAIR) preheated in a heat exchanger (AIRHTR) to reach the gasification temperature. RGibbs reactor (GASIFIER) is chosen to simulate the remaining two processes (gasification and combustion) involved in the gasification of WP–DIS pellets by minimizing Gibbs free energy. Each gasification reaction is restricted by assigning a specific temperature. This allows for reducing the deviation between predicted results and experimental values in terms of syngas composition and LHV [41,51].

The product stream (RAWSYNG) generated from WP–DIS pellets gasification is mixed with char and ash in a mixer (MIXER) to generate a unique flow (MIXFLOW). The char and ash temperature are increased in a heat exchanger (ASCHTR) to equalize that of syngas. Syngas is then cleaned to separate ash and char in an SSplit block (CYCLONE) and cooled down to the ambient temperature of 30 °C in a heat exchanger (COOLER) to meet the engine specifications [43].

The simulation of the ICE is completed by connecting three consecutive blocks of the Aspen Plus library [41,43,49,50]:

- a compressor (COMPR) that models the pressure increase of the incoming air through an isentropic compression;
- an RGibbs reactor (BURN) that simulates the conversion of syngas internal energy to thermal energy through combustion at constant volume by minimizing Gibbs free energy;
- a turbine (TURB) that converts the thermal energy of combustion exhausts (CMBST-GAS) to kinetic energy through an isentropic expansion;
- a heat exchanger (UTIL) where thermal energy present in the stream exiting the turbine (EXITGAS) is extracted by cooling the exhausts to usable temperature (80 °C) [41]. Such thermal energy may be employed in the paper production process or for the district or office heating purposes based on the productivity of the plant.

### 2.1. Modelling Assumptions

The gasification model is developed by applying a non-stoichiometric equilibrium approach based on the minimization of Gibbs free energy as it gives a better agreement with the experimental outcomes than a stoichiometric and kinetic approach, in terms of syngas composition and process performance. Indeed, applying this approach reduces the deviation between numerical and experimental results, significantly increasing the accuracy of the model [52]. The thermodynamic properties of all the conventional components are estimated through the Peng–Robinson equation of state with Boston–Mathias alpha function (PR-BM) [48,53]. Enthalpy and density of non-conventional components (WP–DIS pellets and Ash) are evaluated by Aspen Plus built-in coal models HCOALGEN and DCOALIGT, respectively.

Several simplifying assumptions are considered to avoid the complexity in the gasification and cogeneration model.

Assumptions for gasification [46,52–54]:

- model is zero-dimensional;
- gasification reactions are completed with a steady-state condition;
- pyrolysis is completed instantaneously;

- for a specific zone, the temperature inside the reactor is uniform in all directions (radially and axially) ensuring the isothermal condition;
- hydrodynamic characteristics of the reactor are neglected;
- all the reactions reach an equilibrium condition;
- reaction pathways to form intermediate products are not considered;
- ash, sulphur, nitrogen, and halogen present in WP–DIS pellet are considered nonreactive;
- char is composed of 100% carbon;
- gaseous components show ideal behaviour;
- gasification is completed at ambient pressure;
- tar formation is neglected as commonly considered in numerical modelling of biomass gasification [24,25,41,45]. Indeed, the present analysis aims at evaluating the CHP generation potentiality of WP–DIS pellets, and this simplifying assumption does not significantly affect the goal;
- among the several reactions that occur during biomass gasification, only the six reactions presented in Table 1 with their heat of the reaction [55,56] are considered.

**Table 1.** List of chemical reactions considered for the development of air-gasification of WP–DIS pellets model with their heat of reaction [55,56].

| Reaction ID | Reaction Formula | Reaction Name | Δ (Heat of Reaction), KJ/mol |
|---|---|---|---|
| R1 | $C + H_2O \rightarrow H_2 + CO$ | Water gas | +131.0 |
| R2 | $C + O_2 \rightarrow CO_2$ | Carbon combustion | −393.0 |
| R3 | $C + 2H_2 \rightarrow CH_4$ | Methanation | −74.0 |
| R4 | $CO + H_2O \rightarrow H_2 + CO_2$ | Water gas shift | −41.0 |
| R5 | $C_2H_4 + 3O_2 \rightarrow 2H_2O + 2CO_2$ | Ethene combustion | −964.0 |
| R6 | $2H_2 + O_2 \rightarrow 2H_2O$ | Hydrogen combustion | −242.0 |

Assumptions for the cogeneration model simulation [57]:

- cogeneration process is steady-state;
- potential and kinetic energy changes throughout the system are neglected;
- pressure drops and heat loss from the combustion chamber of the ICE with surroundings are neglected.

### 2.2. Gasification Model Calibration

During gasification modelling, a unique temperature is set for all the reactions mentioned in Table 1. Each reaction has a different equilibrium constant that highly depends on the temperature [58]. Therefore, all the gasification reactions do not reach an equilibrium condition for a specific temperature. Consequently, by using this approach, results predicted in terms of syngas composition and process performances (CCE and CGE) would significantly deviate from the experimental outcomes, reducing the reliability of the model [24].

According to the available literature, the deviation should be lower than ±20% to claim the developed model represents the experimental process [24,41,56]. This condition can be achieved by restricting the equilibrium position of the individual gasification reactions to a specific temperature. Such a temperature can be identified by calibrating the model through experimental results. Consequently, the equilibrium temperature for each reaction differs from the gasification temperature and is calculated using Equation (1):

$$T_{Eqlm} = T_{Gasf} + \Delta T_{Appr}, \tag{1}$$

where, $T_{Eqlm}$ is the equilibrium temperature, $T_{Gasf}$ is the gasification temperature, and $\Delta T_{Appr}$ is a specific value of temperature to which the gasification is restricted.

The difference between the results in terms of individual syngas composition, *LHV* of syngas, *CCE*, and *CGE* predicted through the developed model and experimental outcomes generate a deviation, which is calculated through Equation (2):

$$Deviation\ (\%) = \frac{Simulation\ result - Experimental\ result}{Experimental\ result} \cdot 100, \tag{2}$$

The average deviation of the overall syngas composition is calculated considering the deviations of the individual components according to Equation (3):

$$Average\ Deviation\ (\%) = \frac{1}{n}\sum_{i=1}^{n} |Deviation|, \tag{3}$$

where, $n$ represents the number of syngas components considered during gasification model calibration and validation.

*2.3. Assessment of Process Performance*

Process performances of WP–DIS pellets conversion to syngas through gasification are evaluated by assessing syngas *LHV*, *CGE*, and *CCE* as well as net power ($\dot{P}_{net}$) available from the conversion process.

The *LHV* of syngas depends on its composition and is calculated using Equation (4) [59]:

$$LHV_{syng}(\text{MJ/Nm}^3) = 0.108y_{\text{H}_2} + 0.126y_{\text{CO}} + 0.358y_{\text{CH}_4}, \tag{4}$$

where, $y_{\text{H}_2}$, $y_{\text{CO}}$, and $y_{\text{CH}_4}$ denote the volume fraction of $\text{H}_2$, CO, and $\text{CH}_4$, respectively, present in syngas.

The ratio between the energy flow rate of the syngas and that of the material fed to the gasifier is defined as *CGE* and is evaluated according to Equation (5) [23]:

$$CGE\ (\%) = \frac{LHV_{syng} \cdot \dot{v}_{syng}}{LHV_{fed} \cdot \dot{m}_{fed}} \cdot 100, \tag{5}$$

where, $LHV_{syng}$ and $LHV_{fed}$ represent the *LHV* of syngas in MJ/Nm$^3$ and WP–DIS pellet in MJ/kg, respectively, whereas $\dot{v}_{syng}$ and $\dot{m}_{fed}$ stand for the volumetric flow rate of syngas (Nm$^3$/h) and mass flow rate of WP–DIS pellet (kg/h).

The ratio of carbon flow rate by weight between the product streams (syngas) and reactant (WP–DIS pellet) is the *CCE* that is assessed by Equation (6) [23]:

$$CCE(\%) = \frac{12}{22.4} \cdot \frac{\dot{v}_{syng}}{\dot{m}_{fed} \cdot \text{C}\% \cdot \sum_{i=1}^{5} n_i \cdot y_i} \cdot 100, \tag{6}$$

where, $i$ represents the carbon-containing constituent present in the syngas, C% is the weight fraction of carbon present in the WP–DIS pellet, $n_i$ is the carbon number, and $y_i$ is the fraction of $i$ compound by volume in syngas (i.e.: $\text{C}_1$–$\text{C}_5$).

The difference between primary power available from the generated syngas and that required to complete air preheating is defined $\dot{P}_{net}$ and is stated in Equation (7):

$$\dot{P}_{net} = \dot{P}_{syng} - \dot{P}_{prht}, \tag{7}$$

where, $\dot{P}_{syng}$ is the primary power obtained from syngas and $\dot{P}_{prht}$ denotes the power required to complete air preheating.

The performance of the ICE is assessed by electrical ($\eta_{el}$), thermal ($\eta_{th}$), and system ($\eta_{sys}$) efficiencies, that are calculated according to Equations (8) to (10), respectively.

$$\eta_{el}(\%) = \frac{\dot{N}_{TURB}}{LHV_{Syng} \cdot \dot{v}_{syng}} \cdot 100, \tag{8}$$

$$\eta_{th}(\%) = \frac{\dot{Q}_{EX}}{LHV_{Syng} \cdot \dot{v}_{syng}} \cdot 100, \tag{9}$$

$$\eta_{sys}(\%) = \frac{\dot{N}_{TURB} + \dot{Q}_{EXCH} + \dot{Q}_{EX}}{LHV_{fed} \cdot \dot{m}_{fed} + \dot{Q}_{INPUT}} \cdot 100, \tag{10}$$

where, $\dot{N}_{TURB}$ denotes the effective power obtained from the ICE, $\dot{Q}_{EXCH}$ represents the thermal power accessible during the cooling of syngas before entering the ICE system, $\dot{Q}_{EX}$ is the thermal power that can be recovered by cooling the turbine exhausts to usable temperature (80 °C) [41], and $\dot{Q}_{INPUT}$ is the rate of power associated with RGibbs reactor including air preheating to complete the gasification process.

### 3. Results and Discussion

#### 3.1. Input Parameters

Data related to the operation of the gasifier and the ICE are collected from the literature. The characteristics of the two different WP–DIS pellets (M1 and M2) used in the present research with *LHV* are illustrated in Table 2, whereas the operating conditions of air-gasification of M1 with corresponding syngas properties are shown in Tables 3 and 4. The gasification model is calibrated by applying the operating conditions and syngas composition mentioned in Tables 3 and 4 for the gasification of M1. Since data on the gasification of M2 is not available in the literature, the model is validated against the experimental results related to the gasification of bamboo chips [47]. Indeed, such biomass presents a composition, in terms of ultimate elemental analysis, similar to M1 as highlighted in Table 2. The operating conditions and syngas composition of bamboo chips gasification, used for model validation, are presented in Tables 3 and 4.

**Table 2.** Composition of WP–DIS pellets (M1 and M2) and bamboo chips with *LHV* [11,47].

|  | Properties | M1[d.b.] | M2[d.b.] | Bamboo Chip[d.b.] |
|---|---|---|---|---|
| Proximate analysis (wt.%) | Moisture content | 3.2 | 2.9 | 7.14 |
|  | Volatile matter | 75.2 | 71.4 | 80.06 |
|  | Fixed carbon | 11.7 | 10.2 | 18.33 |
|  | Ash content | 13.1 | 18.4 | 1.61 |
| Ultimate element alanalysis (wt.%) | C | 55.6 | 50.9 | 44.83 |
|  | $H_2$ | 7.6 | 6.9 | 5.96 |
|  | $N_2$ | 0.35 | 0.39 | 0.35 |
|  | S | 0.07 | 0.08 | 0.15 |
|  | $Cl_2$ | 1.56 | 1.55 | - |
|  | $O_2$ | 21.72 | 21.78 | 47.1 |
|  | *LHV* (MJ/kg) | 24.84 | 22.42 | 18.32 |

d.b. = Dry Basis.

CHP generation through the ICE is simulated based on available literature data, presented in Table 5 [41,43,53].

**Table 3.** Operating parameters applied for the gasification of M1 and bamboo chips [11,47].

| Operating Parameters, Units | M1 | Bamboo Chips | | | |
|---|---|---|---|---|---|
| Test conditions | - | I | II | III | IV |
| Temperature, °C | 850 | | 800 | | |
| Pressure, bar | 1.0 | | 1.0 | | |
| ER, (-) | 0.30 | 0.20 | 0.30 | 0.40 | 0.50 |
| Fed flow rate, kg/h | 1.0 | | 1.0 | | |
| Air flow rate, kg/h | 2.45 | 1.11 | 1.66 | 2.21 | 2.77 |

**Table 4.** Syngas properties for gasification of M1 and bamboo chips [11,47].

| Syngas Composition | M1 * | Bamboo Chips ** | | | |
|---|---|---|---|---|---|
| Test conditions | - | I | II | III | IV |
| CO | 6.9 | 24.13 | 18.70 | 11.30 | 6.96 |
| $H_2$ | 3.8 | 16.96 | 11.74 | 7.17 | 3.48 |
| $CO_2$ | 11.8 | 56.30 | 68.70 | 81.30 | 88.70 |
| $CH_4$ | 4.7 | 3.26 | 1.74 | 1.30 | 0.85 |
| $C_2H_4$ | 2.9 | - | - | - | - |
| Syngas *LHV*, ($MJ/Nm^3$) | 5.0 | 5.98 | 4.21 | 2.65 | 1.48 |

* % vol. (Dry basis), ** % mol (Dry and $N_2$ free basis).

**Table 5.** Operating conditions and performance parameters employed for ICE system simulation [41,43,53].

| Operating Parameters, Units | Value |
|---|---|
| Temperature of the syngas entering the ICE, °C | 30.0 |
| Temperature of the air entering the compressor, °C | 20.0 |
| Stoichiometric air ratio used for syngas combustion, (-) | 3.0 |
| Pressure of the air exiting the compressor and entering the combustion chamber, bar | 20.0 |
| Isentropic compression and expansion coefficient, (%) | 90.0 |
| Pressure of the fume exiting the turbine, bar | 1.0 |
| Temperature of the turbine exhausts, °C | 80.0 |

### 3.2. Gasification Model Development

The limit of temperatures estimated to restrict the equilibrium of each gasification reaction together with the fraction of carbon that participates in the reactions to form syngas is illustrated in Table 6.

**Table 6.** Predicted limit of temperature to restrict the equilibrium of gasification reactions and fraction of carbon participating in the reactions.

| Reaction ID. | $\Delta T_{Appr}$ (°C) |
|---|---|
| R1 | −311.83 |
| R2 | −62.62 |
| R3 | −520.73 |
| R4 | 238.51 |
| R5 | 43.94 |
| R6 | 108.24 |
| Fraction of carbon that participates in gasification reactions (%) | 77.98 |

The comparison between syngas composition and *LHV* predicted through the developed model and the experimental outcomes available in the literature [11,47] is depicted in Figure 2 for the calibration step and in Figure 3 for the validation.

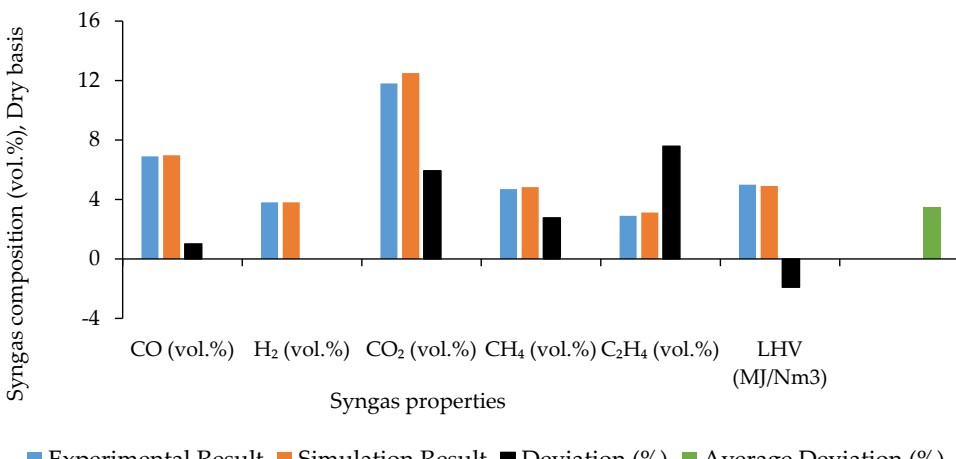

**Figure 2.** Comparison between simulation results predicted during gasification model calibration and experimental outcomes with the corresponding deviation.

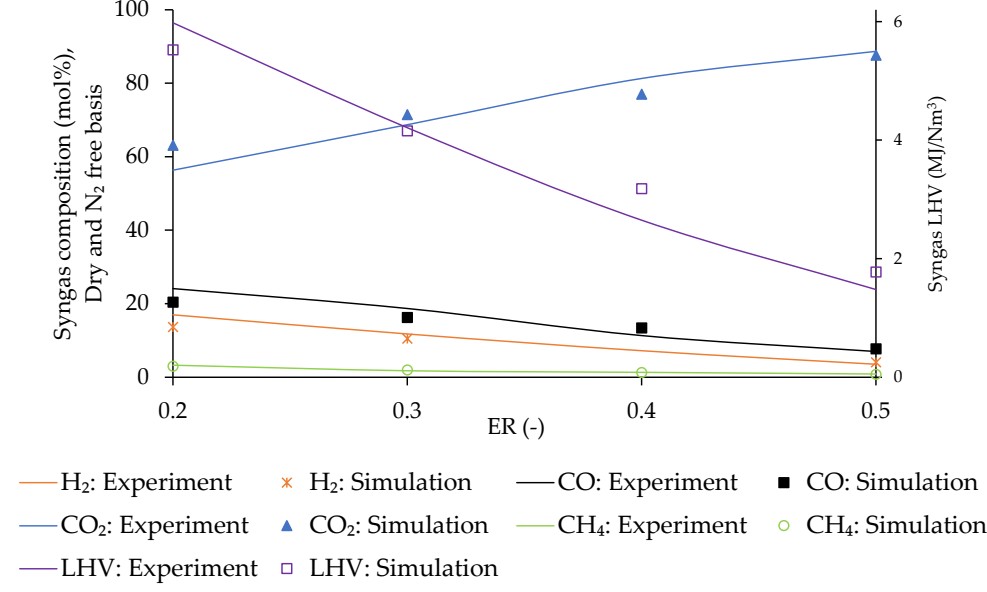

**Figure 3.** Difference between the predicted syngas composition and *LHV* and the experimental data during model validation.

It can be observed that the developed model has a good agreement with the experimental analysis as the average deviation of the syngas composition is 3.46% during model calibration and 11.31% during validation, which satisfies the limit (lower than ±20%) suggested in the literature [24,41,56]. The maximum deviation is obtained for the constituents with the lowest concentration in syngas ($C_2H_4$ during calibration and $CH_4$ for validation). However, the average deviation obtained in the present simulation is substantially lower compared to that of other similar studies that are in the range of 13.70–28.37% [24,41,49,60]. The estimated energy content of the syngas is 1.91%, lower than the experimental one during calibration, due to the over-prediction of $CO_2$, which does not directly affect the *LHV* but creates a dilution effect [54,61,62]. Syngas *LHV* is over- or underpredicted due to $CO_2$ under- or overprediction during model validation.

After validation, the developed model is used to predict the optimum operating conditions of temperature and ER for the conversion of the M2 pellet to syngas through a sensitivity analysis.

### 3.3. Sensitivity Analysis

To optimize the process, the effect of gasification temperature and ER on composition, *LHV*, and density of syngas, *CGE*, *CCE*, and $\dot{P}_{net}$ obtained from thermal treatment of the M2 pellet is analysed by setting a fed flow rate of 1.0 kg/h.

### 3.3.1. Effect of Temperature

Gasification temperature is varied in the range of 700 to 1000 °C at a fixed ER of 0.3 to estimate its optimum value.

The fluctuation of syngas composition with gasification temperature is presented in Figure 4. The variation of *LHV* and density of syngas, *CGE*, *CCE* and $\dot{P}_{net}$ is presented in Figure 5.

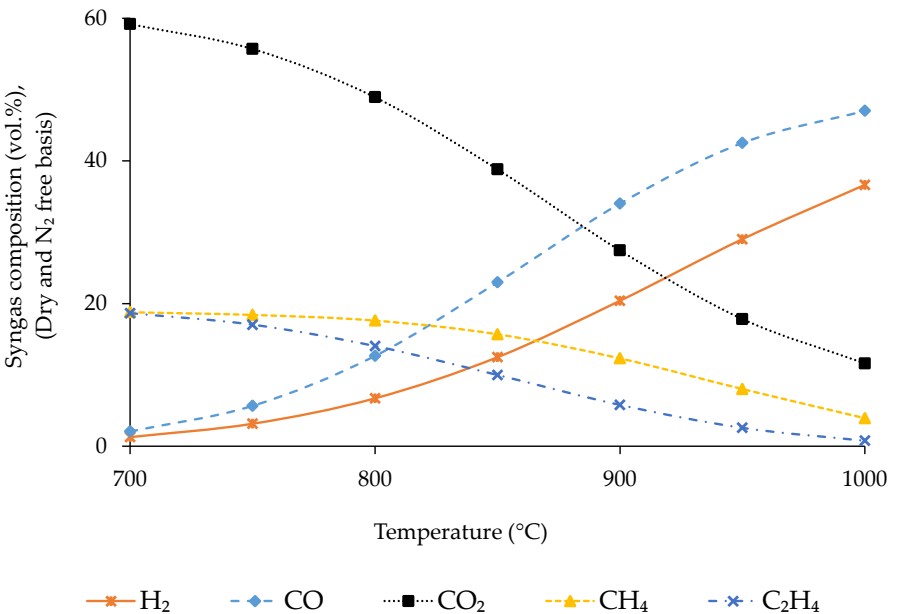

**Figure 4.** Effect of gasification temperature on syngas composition.

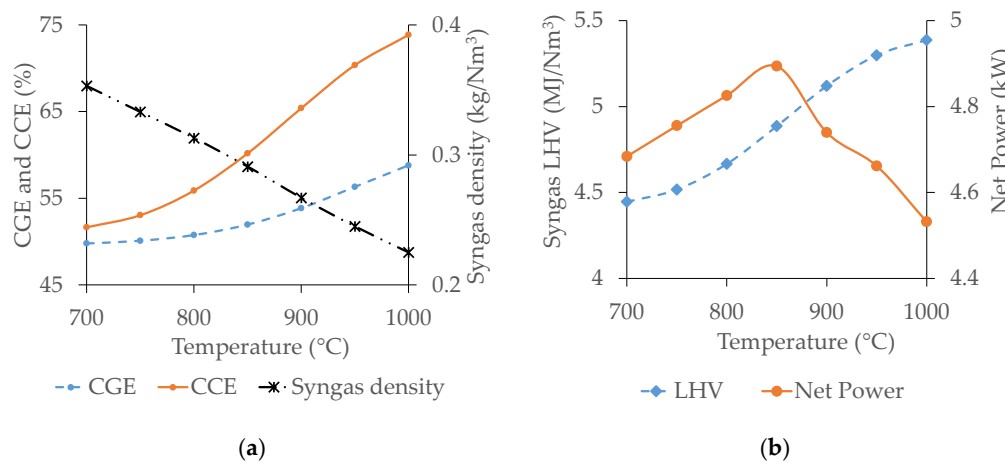

(**a**)                                                                 (**b**)

**Figure 5.** Effect of gasification temperature on *CGE*, *CCE*, and syngas density (**a**) and syngas *LHV* and net power from gasification products (**b**).

The concentration of CO and $H_2$ increases continuously with temperature, whereas that of $CO_2$, $CH_4$, and $C_2H_4$ shows an opposite trend due to the alteration of exothermic and endothermic reaction rates with temperature [49,51,63]. Endothermic reactions (water–gas and water–gas shift reactions in this case) move in the forward direction with temperature,

being responsible for the increment of CO and $H_2$ concentration in syngas. Conversely, exothermic reactions (methanation, ethene formation, and all the combustion reactions) move in the opposite direction with temperature and are responsible for the reduction of $CO_2$, $CH_4$, and $C_2H_4$ concentration in syngas [37,64].

*CGE* and *CCE* increase continuously with temperature whereas syngas density decreases. The energy flow rate exiting from the gasifier associated with the syngas increases with temperature due to the raise of CO and $H_2$ concentration in syngas, as shown in Figure 4. As a consequence, the *CGE* increases as well [23,59]. The raising of carbon fraction in the syngas through CO concentration with temperature is higher compared to the cumulative decrement of the other three carbon-containing species ($CO_2$, $CH_4$, and $C_2H_4$). Additionally, the syngas flow rate increases with temperature, and consequently *CCE* raises [23]. The kinetic energy of molecules present in syngas increases with temperature decreasing its density [65].

The gasification temperature similarly affects syngas *LHV* and $\dot{P}_{net}$. Syngas *LHV* increases continuously with temperature due to the raise of CO and $H_2$ concentration [22,59,66]. Compared to M1 pellets, considering the same operating conditions, the syngas *LHV* is slightly lower due to the higher ash content. The primary power of the product stream increases with temperature as syngas *LHV* raises. However, the thermal power required for air preheating increases simultaneously with temperature. For this reason, the net power available from the gasification products increases continuously with temperature up to 850 °C and afterward, it decreases as the energy required for air preheating (to raise the gasification temperature from 850 to 900 °C and afterward) is higher than the energy gain.

Based on the current analysis, 850 °C appears to be the optimum temperature for gasification of WP–DIS pellets (M2). Indeed, increasing the gasification temperature above this value does not appear to be convenient in terms of the net energy that can be recovered from the gasification products.

### 3.3.2. Effect of ER

The variation in composition, *LHV*, and density of syngas, *CGE*, *CCE* and $\dot{P}_{net}$ is assessed by varying the ER from 0.1 to 0.4 at the predicted optimum temperature of 850 °C. The results are presented in Figures 6 and 7. Regarding syngas composition, the concentration of $CO_2$, $H_2$, and CO increases with ER whereas that of $CH_4$ and $C_2H_4$ decreases due to the movement of oxidation reaction to the forward direction with the raise of $O_2$ concentration inside the gasifier, as explained by Le Chatelier's principle [54].

*CGE* continuously decreases with ER, whereas *CCE* and syngas density experience an opposite trend. The concentration of $C_2H_4$ and $CH_4$ decreases by 76.8% and 46.9%, respectively, with the increase of ER within the tested range. Conversely, the concentration of CO increases by 24.2% and that of $H_2$ by 53.8%, as clearly presented in Figure 6. The contribution of $CH_4$ fraction to the energy content of syngas is almost three times compared to that of CO and $H_2$ [22,59,66]. Consequently, the *CGE* decreases continuously with ER. Carbon transformation from input biomass to the gasification product increases due to the raise of oxidation reactions with ER. Additionally, the syngas flow rate increases with ER, and consequently *CCE* raises continuously [23]. The concentration of $N_2$ inside the gasifier raises with ER being responsible for the reduction of molecular movement in the reacting medium due to its inert nature. This causes an increase in the syngas density [67].

Syngas *LHV* decreases with ER due to the increase of $N_2$ volume inside the reactor, which causes a dilution effect [54,61,62]. $\dot{P}_{net}$ obtained from syngas decreases with ER due to the reduction of *LHV*. At the same time, the incoming air flow rate increases with ER requiring more thermal energy for air preheating, further decreasing the available $\dot{P}_{net}$.

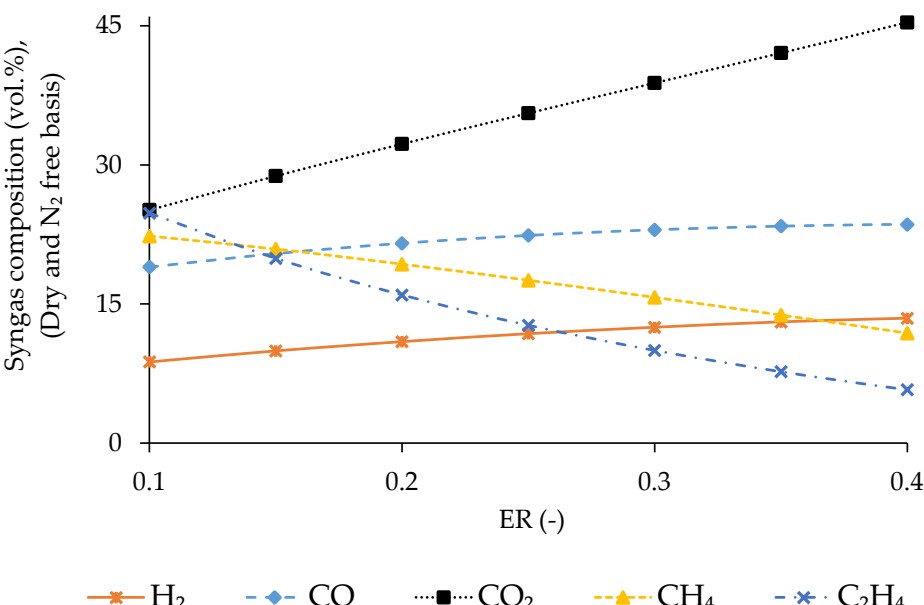

**Figure 6.** Fluctuation of syngas composition with ER.

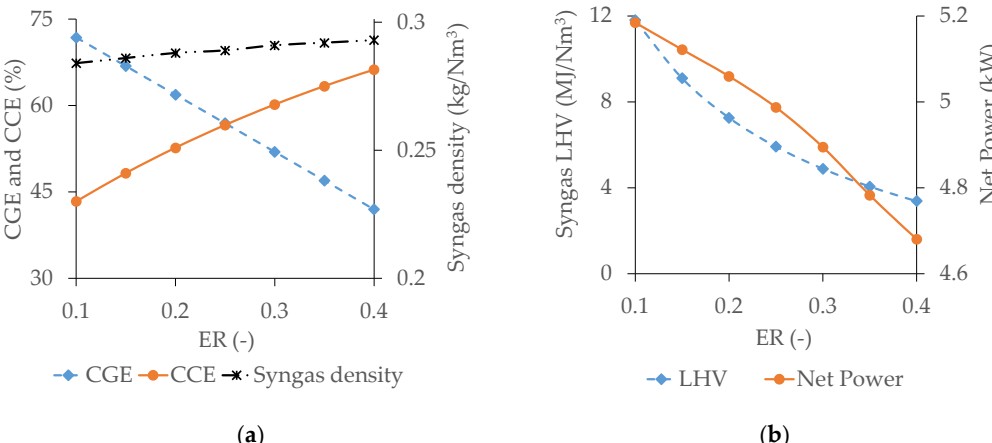

**Figure 7.** Effect of ER on *CGE*, *CCE*, and syngas density (**a**) and syngas *LHV* and net power from products (**b**).

The estimated optimum ER appears to be at least 0.2. Indeed, although syngas *LHV*, *CGE*, and $\dot{P}_{net}$ are higher at ER 0.15, thermal treatment of biomass by applying an ER lower than 0.2 leads to pyrolysis rather than gasification, which generates more tar, char, ash, and other impurities [49,68]. The removal of all the impurities from syngas, including tar content, is required before its use in an ICE, to ensure a high conversion efficiency and engine lifetime. The cleaning process for syngas generated at ER 0.15 would be more costly compared to the reduction of energy content obtained at 0.2 [69].

The fluctuation of composition and *LHV* of syngas, *CCE*, and *CGE* predicted in the current analysis with temperature and ER is in accordance with the studies on syngas generation through thermal treatment of biomass available in the literature [24,25,33,41,42,49,51,54,60–62,68,70,71].

### 3.3.3. Cogeneration Process Performances

The exhaust gas of the ICE fuelled with syngas from gasification of WP–DIS pellets contains $CO_2$, $NO_X$, and HCl. The variation of emission profiles with the analysed operating parameters is presented in Figure 8.

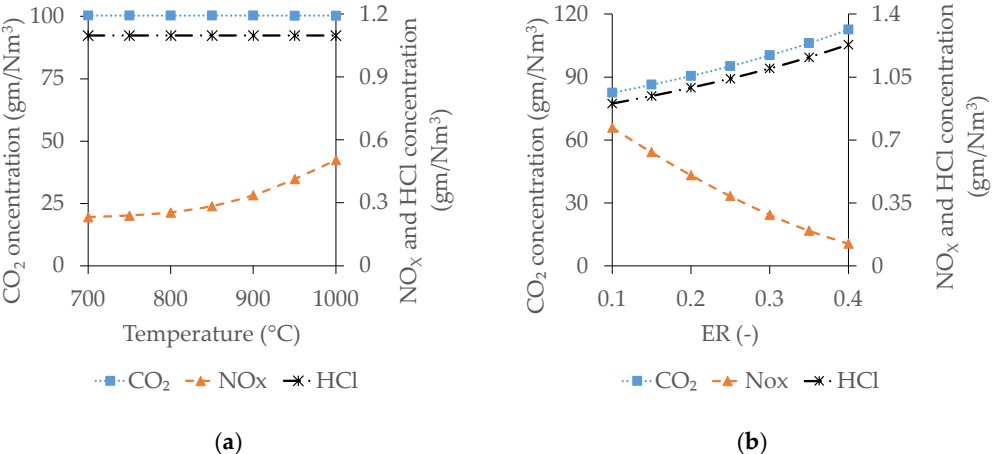

**Figure 8.** Variation of ICE emission profile with gasification operating parameters of temperature (**a**) and ER (**b**).

As shown in Figure 8, the concentration of $NO_X$ increases with the temperature as the $NO_X$ formation reaction is endothermic and moves in the forward direction when the temperature increases [67]. The concentration of the remaining two constituents ($CO_2$ and HCl) in the exhaust does not significantly change with gasification temperature. Analysing the effect of ER, the concentration of $NO_X$ decreases with ER whereas that of $CO_2$ and HCl increases. With the raise of ER, combustion reactions move into forward direction, and consequently the concentration of $CO_2$ and HCl increases. At the same time, the raising of $N_2$ concentration inside the gasifier with ER creates a dilution effect, reducing the temperature, and, thus, $NO_X$ formation decreases continuously [54,61,62].

The estimated potentiality of CHP generation from the WP–DIS is 1.14 kWh/kg as DS of electrical energy and 1.80 kWh/kg as DS of thermal energy.

A summary of the predicted *CGE*, $\eta_{el}$, $\eta_{th}$, and $\eta_{sys}$ obtained in the present study, together with their comparison with relevant analysis available in the literature, is illustrated in Table 7.

**Table 7.** Comparison of CGE, $\eta_{el}$, $\eta_{th}$, and $\eta_{sys}$ predicted in the present analysis with similar studies available in the literature.

| Biomass | *CGE* | $\eta_{el}$ | $\eta_{th}$ | $\eta_{sys}$ | Ref. |
|---|---|---|---|---|---|
| WP–DIS pellet (M2) | 61.90 | 18.32 | 46.86 | 73.87 | Present study |
| SS | n.r. | 29.20 | 45.92 | 53.10 | [41] |
|  | 58.1 | 19.3 | 48.7 | n.r. | [42] |
| Hazelnut shell | 60.0 | 30.0 | n.r. | 64.0 | [43] |
| Olive pruning | 35.0 | 26.0 | n.r. | 41.0 | [43] |
| MSW | 59.0 | 19.1 | 20.0 | 40.1 | [52] |
| Olive kernel | 70.0 | n.r. | 33.5 | n.r. | [53] |
| Wood | n.r. | 27.0 | 40.0 | 67.0 | [70] |
|  | 84.0 | 27.0 | 39.0 | 66.0 | [71] |

n.r. = Not reported.

Based on the data related to WP and DIS generated in the EU in 2019, it is estimated that 32,950–35,700 GWh of electrical and 52,190–56,100 GWh of thermal energy could be produced based on the DIS generation in the range of 20–40 wt% of WP [4,12]. By considering the emission factors for electricity consumption and natural gas combustion, the avoided $CO_2$ emissions resulting from using the producible electrical and thermal energy would be in the ranges of 12998–15164 and 11781–13744 Mt of $CO_2$ per year, respectively.

Finally, by considering the estimated CHP generation potentiality from WP–DIS in 2019, it is estimated that between 25 and 28% and from 44 to 48% of the electrical and

thermal energy demand, respectively, of the pulp and paper manufacturing sector, could be fulfilled in the EU.

## 4. Conclusions

An energy-efficient solution for the paper industry, based on energy recovery from waste paper and deinking sludge, is presented in this work. The proposed solution consists of CHP generation from waste paper and deinking sludge blends through gasification in combination with an internal combustion engine. A gasification model is developed considering the experimental results on gasification of waste paper and deinking sludge blends and bamboo chips available in the literature.

Sensitivity analyses were performed to predict the optimum operating conditions of temperature and equivalence ratio by assessing their effect on syngas composition, lower heating value, cold gas efficiency, carbon conversion efficiency, and net power obtained from the conversion process. Temperature raising has a positive impact on the process as it increases the syngas lower heating value, cold gas efficiency, carbon conversion efficiency, and net available power whereas the equivalence ratio has a reverse effect.

Estimating CHP generation potentiality from waste paper and deinking sludge in the EU in 2019 through the proposed system allows us to highlight that:

- between 25 and 28% of the electrical and between 44 and 48% of thermal energy demand in the pulp and paper manufacturing sector could be supplied;
- this would allow saving between 24.8 and 28.9 Gt of $CO_2$ per year.

Therefore, the proposed system can significantly contribute to reducing greenhouse gas emissions caused by the current management practices used for waste disposal in the paper recycling industry as well as by its consumption of electrical and thermal energy, which comes from fossil fuels. This, in accordance with the goals of the EU Green Deal 2021, would also reduce greenhouse gas emissions and increase the renewable energy generation in this sector [72].

In order to better analyse the environmental benefits of the proposed system, a life cycle assessment should be carried out as future development of this study.

**Author Contributions:** Conceptualization, S.D.F. and M.R.U.; methodology, S.D.F. and M.R.U.; software, S.D.F. and M.R.U.; validation, S.D.F. and M.R.U.; formal analysis, M.R.U.; investigation, S.D.F. and M.R.U.; resources, M.R.U.; data curation, S.D.F. and M.R.U.; writing—original draft preparation, M.R.U.; writing—review and editing, S.D.F.; visualization, M.R.U.; supervision, S.D.F. All authors have read and agreed to the published version of the manuscript.

**Funding:** This research received no external funding.

**Conflicts of Interest:** The authors declare no conflict of interest.

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
