# Peer review of "Energy Recovery from Waste Paper and Deinking Sludge to Support the Demand of the Paper Industry: A Numerical Analysis"

_sustainability, doi:10.3390/su14084669_

Round 1
Reviewer 1 Report
Journal: Sustainability
Comments on the manuscript entitled “Title: Recovery of wastepaper and deinking sludge to improve the energy efficiency of the paper industry” (Manuscript No. Sustainability-1664692)
The article is not suitable for publication in its present form. It needs a major revision. Below are my comments:
Some specific comments:
- Abstract: It is suggested to add some background with few objectives and possible applications of this study and highlight the novelty of this work. The abstract only contains some parameters without any process conditions or key values from results, which is insufficient to delineate the whole pictures of contribution and possible application of this study.
- Revise keywords add more specific and novel keywords with broader meanings (5-7 words).
- Add some references in the Introduction section to strengthen the literature review and compare the greenhouse gas emissions:
- Yaqoob H, Teoh YH, Jamil MA, Gulzar M. Potential of tire pyrolysis oil as an alternate fuel for diesel engines: A review. Journal of the Energy Institute 2021;96:1–17. doi:https://doi.org/10.1016/j.joei.2021.03.002.
- Murugesan, A.; Umarani, C.; Subramanian, R.; Nedunchezhian, N. Bio-diesel as an alternative fuel for diesel engines—A review. Renew. Sustain. Energy Rev. 2009, 13, 653–662.
- Ayanoglu, R. Yumrutas¸ , Production of gasoline and diesel like fuels from
waste tire oil by using catalytic pyrolysis, Energy 103 (2016) 456e468,
https://doi.org/10.1016/j.energy.2016.02.155.
- The introduction is lacks sufficient background information, which is unable to give the reader detailed background knowledge and possible wide application of this study. Research gaps should be highlighted more clearly and future applications of this study should be added.
- The detailed specification of the reactors is missing.
- The standard of emission analysis is missing in the manuscript.
- Redraw Figure No. 2 for better visualization.
- Elaborate the effects of physicochemical properties on the performance and emission characteristics.
- Between the lack of methodology and limited novelty, the authors have a significant amount of work to do to bring this paper up to a publishable standard.
Author Response
Dear Reviewer 1
We wish to thank the reviewer for his valuable comments which have helped us to improve the quality of this manuscript. We have carried out a revision of the manuscript by using “Track Changes” function of MS Word to address the issues raised by the reviewers. Please find the answers point by point in red color attached herewith.

Reviewer 2 Report
After reviewing carefully, the reviewer found that the paper is well written. However, a minor revision is required to improve the quality of the manuscript further.
1. The abstract needs style and form corrections since the ideas sometimes seems unclear. Please try to smooth the connections in between and organize them logically, to make them sound as a solid single paragraph.
2. Add paper organization at the end of the introduction.
3. Improve the quality of Figure 4, 5, 6, 7. Add better resolution images.
Author Response
Dear Reviewer 2
We wish to thank the reviewer for his valuable comments which have helped us to improve the quality of this manuscript. We have carried out a revision of the manuscript by using “Track Changes” function of MS Word to address the issues raised by the reviewers. Please find the answers point by point in red color attached herewith.

Reviewer 3 Report
This manuscript deals with research on the management of different wastes produced from the paper industry and increasing its energy efficiency. I have thoroughly checked the manuscript. However, I have a bit of concern regarding the fluency and English language. The manuscript needs a thorough English check and arrangement of paragraphs as per the proposed hypothesis. Many short paragraphs don’t make any sense and need merging with other paragraphs. Introduction used so many references which is pointless. I recommend major revision.
My specific comments are:
- Abstract line 10: Add space between “WastePaper”.
- Line 10: Change “reduce” to “reducing”.
- Line 11: change to “high energy demand”.
- Write full form of EU.
- Move “for the first time” to the end of this sentence.
- Provide a concluding statement at the end of the abstract indicating the novelty of this study and its applicability in real-time.
- The spacing problem exists throughout the manuscript. E.g. GreenHouse line 24. Correct other spacing problems in the whole manuscript.
- Do not start the paragraphs or sentences with the names of the authors. E.g. line 104.
- Line 118: Provide make, version, city, country of Aspen Plus software with brackets.
- Line 130, too short paragraph. Merge it with the next one.
- Line 162: describe in Ref. [25]? This is not a standard practice of scientific writing.
- Table 1: 0.5 O2 is not standard practice for writing chemical equations. Either change it to ½ O2 or 2H2+O2-->2H2O+energy. I would insist on using integers for balancing equations. Or do authors have a valid reason for using non-integers numbers? Also, the reactions do not show that energy is liberated, add it.
- Change “Ultimate analysis” to “Ultimate elemental analysis”.
- Either use “&” or “and”.
- The resolution and quality of the figures are distorted. I recommend authors use PNG files without distorting the original resolution and aspect ratio.
- Improve the histogram of bar/line figures and add error bars (if applicable).
- The line “483: Change “will be” to “should be”.
- Avoid outdated references.
Author Response
Dear Reviewer 3
We wish to thank the reviewer for his valuable comments which have helped us to improve the quality of this manuscript. We have carried out a revision of the manuscript by using “Track Changes” function of MS Word to address the issues raised by the reviewers. Please find the answers point by point in red color attached herewith.

Reviewer 4 Report
Dear Authors,
Your work presents interesting issue, but I have some comment and suggestions:
- Please improve the qality of figures. Now, they are indistinct.
- The values of x axis must not start at for example 700. Please correct it and add 0 in alle figures.
- L: 162 - please remove word "Ref.". It is unnecessary.
Author Response
Dear Reviewer 4
We wish to thank the reviewer for his valuable comments which have helped us to improve the quality of this manuscript. We have carried out a revision of the manuscript by using “Track Changes” function of MS Word to address the issues raised by the reviewers. Please find the answers point by point in red color attached herewith.

Round 2
Reviewer 3 Report
The authors have revised the manuscript as per my suggestions and answered all my queries well. I do not see any reason hindering the acceptance of this manuscript. I suggest acceptance in current form. Thank you.